# Polygenic adaptation on height is overestimated due to uncorrected stratification in genome-wide association studies

Mashaal Sohail[1,2,3†§*], Robert M Maier[3,4,5†*], Andrea Ganna[3,4,5,6,7], Alex Bloemendal[3,4,5], Alicia R Martin[3,4,5], Michael C Turchin[8,9], Charleston WK Chiang[10], Joel Hirschhorn[3,11,12], Mark J Daly[3,4,5,7], Nick Patterson[3,13], Benjamin Neale[3,4,5‡*], Iain Mathieson[14‡*], David Reich[3,13,15‡*], Shamil R Sunyaev[2,3,16‡*]

[1]Division of Genetics, Department of Medicine, Brigham and Women's Hospital and Harvard Medical School, Boston, United States; [2]Department of Biomedical Informatics, Harvard Medical School, Boston, United States; [3]Program in Medical and Population Genetics, Broad Institute of MIT and Harvard, Cambridge, United States; [4]Stanley Center for Psychiatric Research, Broad Institute of MIT and Harvard, Cambridge, United States; [5]Analytical and Translational Genetics Unit, Massachusetts General Hospital, Boston, United States; [6]Department of Medical Epidemiology and Biostatistics, Karolinska Institutet, Stockholm, Sweden; [7]Institute for Molecular Medicine Finland, University of Helsinki, Helsinki, Finland; [8]Center for Computational Molecular Biology, Brown University, Providence, United States; [9]Department of Ecology and Evolutionary Biology, Brown University, Providence, United States; [10]Department of Preventive Medicine, Center for Genetic Epidemiology, Keck School of Medicine, University of Southern California, Los Angeles, United States; [11]Departments of Pediatrics and Genetics, Harvard Medical School, Boston, United States; [12]Division of Endocrinology and Center for Basic and Translational Obesity Research, Boston Children's Hospital, Boston, United States; [13]Department of Genetics, Harvard Medical School, Boston, United States; [14]Department of Genetics, Perelman School of Medicine, University of Pennsylvania, Philadelphia, United States; [15]Howard Hughes Medical Institute, Harvard Medical School, Boston, United States; [16]Division of Genetics, Department of Medicine, Brigham and Women's Hospital and Harvard Medical School, Boston, United States

*For correspondence:
mashaal33@gmail.com (MS);
rmaier@broadinstitute.org (RMM);
bneale@broadinstitute.org (BN);
mathi@pennmedicine.upenn.edu
(IM);
reich@genetics.med.harvard.edu
(DR);
ssunyaev@rics.bwh.harvard.edu
(SRS)

†These authors contributed equally to this work
‡These authors also contributed equally to this work

Present address: §National Laboratory of Genomics for Biodiversity (UGA-LANGEBIO), Cinvestav, Irapuato, Mexico

**Abstract** Genetic predictions of height differ among human populations and these differences have been interpreted as evidence of polygenic adaptation. These differences were first detected using SNPs genome-wide significantly associated with height, and shown to grow stronger when large numbers of sub-significant SNPs were included, leading to excitement about the prospect of analyzing large fractions of the genome to detect polygenic adaptation for multiple traits. Previous studies of height have been based on SNP effect size measurements in the GIANT Consortium meta-analysis. Here we repeat the analyses in the UK Biobank, a much more homogeneously designed study. We show that polygenic adaptation signals based on large numbers of SNPs below genome-wide significance are extremely sensitive to biases due to uncorrected population stratification. More generally, our results imply that typical constructions of polygenic scores are

sensitive to population stratification and that population-level differences should be interpreted with caution.

**Editorial note:** This article has been through an editorial process in which the authors decide how to respond to the issues raised during peer review. The Reviewing Editor's assessment is that all the issues have been addressed (see decision letter).

DOI: https://doi.org/10.7554/eLife.39702.001

## Introduction

Most human complex traits are highly polygenic (*Yang et al., 2010*; *Boyle et al., 2017*). For example, height has been estimated to be modulated by as much as 4% of human allelic variation (*Boyle et al., 2017*; *Zeng et al., 2018*). Polygenic traits are expected to evolve differently from monogenic ones, through slight but coordinated shifts in the frequencies of a large numbers of alleles, each with mostly small effect. In recent years, multiple methods have sought to detect selection on polygenic traits by evaluating whether shifts in the frequency of trait-associated alleles are correlated with the signed effects of the alleles estimated by genome-wide association studies (GWAS) (*Turchin et al., 2012*; *Berg and Coop, 2014*; *Mathieson et al., 2015*; *Robinson et al., 2015*; *Berg et al., 2017*; *Racimo et al., 2018*; *Guo et al., 2018*).

Here we focus on a series of recent studies—some involving co-authors of the present manuscript—that have reported evidence of polygenic adaptation at alleles associated with height in Europeans. One set of studies observed that height-increasing alleles are systematically elevated in frequency in northern compared to southern European populations, a result that has subsequently been extended to ancient DNA (*Turchin et al., 2012*; *Berg and Coop, 2014*; *Mathieson et al., 2015*; *Robinson et al., 2015*; *Berg et al., 2017*; *Racimo et al., 2018*; *Guo et al., 2018*; *Simonti et al., 2017*). Another study using a very different methodology (singleton density scores, SDS) found that height-increasing alleles have systematically more recent coalescence times in the United Kingdom (UK) consistent with selection for increased height in the last few thousand years (*Field et al., 2016a*). In the present work, we assess polygenic adaptation on human height as a particular case of the effects that uncorrected population structure in GWAS can have on studies of complex traits.

Most of these previous studies have been based on SNP associations and effect sizes (summary statistics) reported by the GIANT Consortium, which most recently combined 79 individual GWAS through meta-analysis, including a total of 253,288 individuals (*Lango Allen et al., 2010*; *Wood et al., 2014*). Here, we show that the selection effects described in these studies are severely attenuated and in some cases no longer significant when using summary statistics derived from the UK Biobank, an independent and larger study that includes 336,474 genetically unrelated individuals who derive their recent ancestry almost entirely from the British Isles (identified as 'white British ancestry' by the UK Biobank) (*Supplementary file 1*). The UK Biobank analysis is based on a single cohort drawn from a relatively homogeneous population enabling better control of population stratification. Both datasets have high concordance even for low P value SNPs which do not reach genome-wide significance (*Figure 1—figure supplement 1*; genetic correlation between the two height studies is 0.94 [se = 0.0078]). Despite this concordance, we observe that small but systematic biases lead to the two datasets yielding qualitatively different conclusions with respect to signals of polygenic adaptation.

## Results

### Discrepancies in GWAS: population-level differences in height

To study population level differences among ancient and present-day European samples, we began by estimating 'polygenic height scores' as sums of allele frequencies at independent SNPs weighted by their effect sizes from GIANT. We used a set of different significance thresholds and strategies to correct for linkage disequilibrium as employed by previous studies, and replicated their signals for significant differences in genetic height across populations (*Turchin et al., 2012*; *Berg and Coop, 2014*; *Mathieson et al., 2015*; *Robinson et al., 2015*; *Berg et al., 2017*; *Racimo et al., 2018*; *Guo et al., 2018*; *Simonti et al., 2017*) (*Figure 1a*, *Figure 1—figure supplement 2*). We then

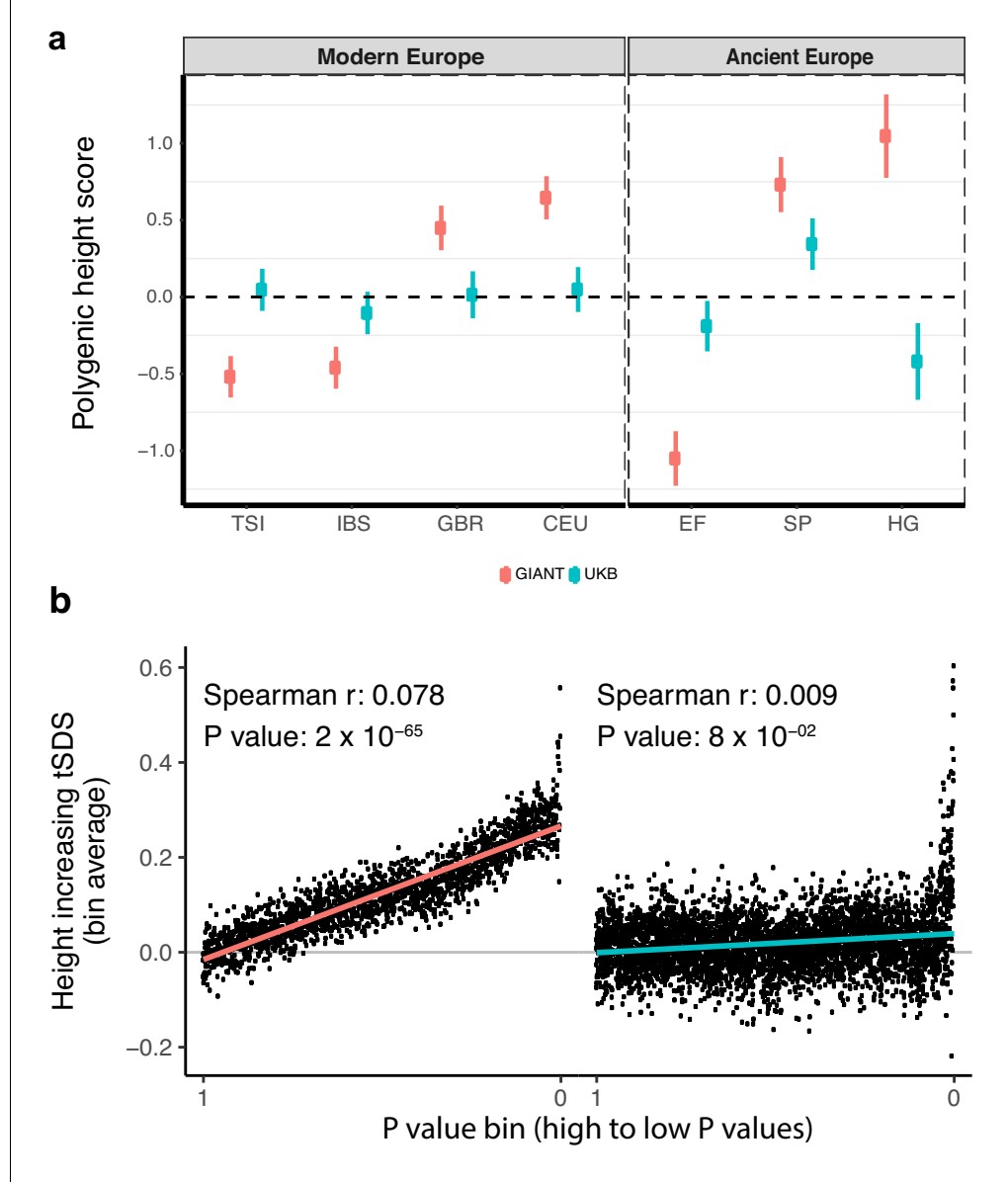

**Figure 1.** Polygenic height scores and tSDS scores based on GIANT and UK Biobank GWAS. (a) Polygenic scores in present-day and ancient European populations are shown, centered by the average score across populations and standardized by the square root of the additive variance. Independent SNPs for the polygenic score from both GIANT (*red*) and the UK Biobank [UKB] (*blue*) were selected by picking the SNP with the lowest P value in each of 1700 independent LD blocks similarly to refs. (**Berg et al., 2017**; **Racimo et al., 2018**) (see Materials and methods). Present-day populations are shown from Northern Europe (CEU, GBR) and Southern Europe (IBS, TSI) from the 1000 genomes project; Ancient populations are shown in three meta-populations (HG = Hunter Gatherer (n = 162 individuals), EF = Early Farmer (n = 485 individuals), and SP = Steppe Ancestry (n = 465 individuals)) (see **Supplementary file 2**). Error bars are drawn at 95% credible intervals. See **Figure 1—figure supplement 1** for analyses of concordance of effect size estimates between GIANT and UKB. See **Figure 1—figure supplements 2–6** for polygenic height scores computed using other linkage disequilibrium pruning procedures, significance thresholds, summary statistics and populations. (b) tSDS for height-increasing allele in GIANT (left) and UK Biobank (right). The tSDS method was applied using pre-computed Singleton Density Scores for 4,451,435 autosomal SNPs obtained from 3195 individuals from the UK10K project (**Field et al., 2016a**; **Field et al., 2016b**) for SNPs associated with height in GIANT and the UK Biobank. SNPs were ordered by GWAS P value and grouped into bins of 1000 SNPs each. The mean tSDS score within each P value bin is shown on the y-axis. The Spearman correlation coefficient between the tSDS scores and GWAS P values, as well as the correlation standard errors and P values, were computed on the un-binned data. The gray line indicates the null-

*Figure 1 continued on next page*

*Figure 1 continued*

expectation, and the colored lines are the linear regression fit. The correlation is significant for GIANT (Spearman r = 0.078, p=1.55×10$^{-65}$) but not for UK Biobank (Spearman r = −0.009, p=0.077). See **Figure 1—source data 1** for figure data.

DOI: https://doi.org/10.7554/eLife.39702.002

The following source data and figure supplements are available for figure 1:

**Source data 1.** Polygenic height scores and tSDS scores based on GIANT and UK Biobank GWAS.

DOI: https://doi.org/10.7554/eLife.39702.009

**Figure supplement 1.** Beta concordance between GIANT and UK Biobank by P value bin.

DOI: https://doi.org/10.7554/eLife.39702.003

**Figure supplement 2.** Polygenic height scores based on GIANT and UK Biobank GWAS for clumped SNPs in present-day and ancient Europeans.

DOI: https://doi.org/10.7554/eLife.39702.004

**Figure supplement 3.** Polygenic height scores in 1000 genomes European populations using clumped SNPs and effect sizes from different summary statistics.

DOI: https://doi.org/10.7554/eLife.39702.005

**Figure supplement 4.** Polygenic height scores in 1000 Genomes Project European populations using ~1700 independent SNPs and effect sizes from different summary statistics.

DOI: https://doi.org/10.7554/eLife.39702.006

**Figure supplement 5.** Polygenic height scores in ancient populations using ~1700 independent SNPs and effect sizes from different summary statistics.

DOI: https://doi.org/10.7554/eLife.39702.007

**Figure supplement 6.** Polygenic height scores in ancient and global modern populations using three different GWAS.

DOI: https://doi.org/10.7554/eLife.39702.008

repeated the analysis using summary statistics from a GWAS for height in the UK Biobank restricting to individuals of British Isles ancestry (hereafter referred to as the 'white British' (WB) subset) and correcting for population stratification based on the first ten principal components (UK Biobank [UKB]; also referred to as 'UKB Neale' in the supplementary figures) (*Churchhouse et al., 2017*). This analysis resulted in a dramatic attenuation of differences in polygenic height scores (*Figure 1a*, *Figure 1—figure supplements 2–4*). The differences between ancient European populations also greatly attenuated (*Figure 1a*, *Figure 1—figure supplement 5*). Strikingly, the ordering of the scores for populations also changed depending on which GWAS was used to estimate genetic height both within Europe (*Figure 1a*, *Figure 1—figure supplements 2–5*) and globally (*Figure 1—figure supplement 6*), consistent with reports from a recent simulation study (*Martin et al., 2017*). The height scores were qualitatively similar only when we restricted to independent genome-wide significant SNPs in GIANT and the UK Biobank (p<5×10$^{-8}$) (*Figure 1—figure supplement 2b*). This replicates the originally reported significant north-south difference in the allele frequency of the height-increasing allele (*Turchin et al., 2012*) or in genetic height (*Berg and Coop, 2014*) across Europe, as well as the finding of greater genetic height in ancient European steppe pastoralists than in ancient European farmers (*Mathieson et al., 2015*), although the signals are attenuated even here. Our observations suggest that tests of polygenic adaptation based on genome-wide significant SNPs are relatively consistent across different GWAS (*Figure 1—figure supplement 2b*) and that our concern is primarily directed towards the use of sub-significant SNPs in polygenic scores (*Figure 1a*, *Figure 1—figure supplement 2a*).

## Discrepancies in GWAS: height evolution within a single population

Next, we assessed if an independent measure, the 'singleton density score (SDS)', which uses a coalescent approach to infer adaptation within a population, is equally as susceptible to biases in GWAS (*Field et al., 2016a*; *Field et al., 2016b*). SDS can be combined with GWAS effect size estimates to infer polygenic adaptation on complex traits (generating a 'tSDS score' by aligning the SDS sign to the trait-increasing allele). A tSDS score larger than zero for height-increasing alleles implies that these alleles have been increasing in frequency in a population over time due to natural selection. We replicate the original finding that SDS scores of the height-increasing allele computed

in the UK population (using the UK10K dataset) increase with stronger association of the alleles to height as inferred by GIANT (*Field et al., 2016a*) across the entire P value spectrum (Spearman's ρ = 0.078, p=1.55×10$^{-65}$, *Figure 1b*). However, we observed that this signal of polygenic adaptation in the UK, measured using a Spearman correlation across all GWAS SNPs, disappeared when we used the UK Biobank height effect size estimates (ρ = 0.009, p=0.077, *Figure 1b*). These observations suggest that concerns about sub-significant SNPs should not only be directed towards population-level differences using polygenic scores but also to analyses of adaptation within a single population.

## Population structure underlying discrepancies in GWAS

### Discrepancies between GIANT and UK biobank

We propose that the qualitative difference between the polygenic adaptation signals in GIANT and the UK Biobank is due to the cumulative effect of subtle biases in each of the SNPs estimated in GIANT. This bias can arise due to incomplete control of the population structure in GWAS (*Novembre and Barton, 2018*). For example, if height were differentiated along a north-south axis because of differences in environment, any variant that is differentiated in frequency along the same axis would have an artificially large effect size estimated in the GWAS. Population structure is substantially less well controlled for in the GIANT study than in the UK Biobank study. This is both because the GIANT study population is more heterogeneous than that in the UK Biobank, and because population structure in the GIANT meta-analysis may not have been well controlled in some component cohorts due to their relatively small sizes (i.e., the ability to detect and correct population structure is dependent on sample size (*Patterson et al., 2006*; *Price et al., 2006*). The GIANT meta-analysis also found that such stratification effects worsen as SNPs below genome-wide significance are used to estimate height scores (*Wood et al., 2014*), consistent with our finding that the differences in genetic height among populations increase when including these SNPs.

We obtained direct confirmation that population structure is more correlated with effect size estimates in GIANT than to those in the UK Biobank. *Figure 2a* shows that the effect sizes estimated in GIANT, in contrast to those in the UKB, are highly correlated with the SNP loadings of several principal components of population structure (PC loadings). We also find that the UK Biobank estimates including individuals of diverse ancestry and not correcting for population structure (UKB all no PCs) show the same stratification effects as GIANT (*Figure 2—figure supplements 1–3*). Further, in line with our intuition regarding the effects of residual stratification on GWAS effect size estimates, we find that alleles that are more common in the Great Britain population (1000 genomes GBR) than in the Tuscan population from Italy (1000 genomes TSI) tend to be preferentially estimated as height-increasing according to the GIANT study but not according to the UKB study (*Figure 2c*, *Figure 2—figure supplements 2–3*).

### Effect size estimates from previously published family-based height GWAS

We analyzed previously released family-based effect size estimates based on an approach of *Robinson et al. (2015)* (NG2015 sibs). Surprisingly, we found that while these summary statistics produced significant polygenic adaptation signals, they were also correlated with PC loadings as well as with GBR-TSI allele frequency differences (*Figure 2—figure supplements 1–3*). This suggests that these estimates are also affected by population structure despite being computed within families and, therefore, in principle, robust to structure. Our own family-based estimates in the UK Biobank (UKB sibs all, UKB sibs WB) appear unconfounded and do not produce significant adaptation signals across the spectrum of associated SNPs (*Figure 2—figure supplements 1–3*). The residual structure in the original NG2015 sibs dataset is likely to reflect a technical artifact (personal communication from Peter Visscher, and note on their website [*Program in Complex Trait Genomics, 2018*]). Berg and colleagues (*Berg et al., 2019*) show that the updated NG2015 sibs summary statistics (posted in the public domain [*Program in Complex Trait Genomics, 2018*] in November 2018 during the revision of this manuscript) do not show significant signals of polygenic adaptation using either polygenic score differences in Europe or the tSDS metric in the UK.

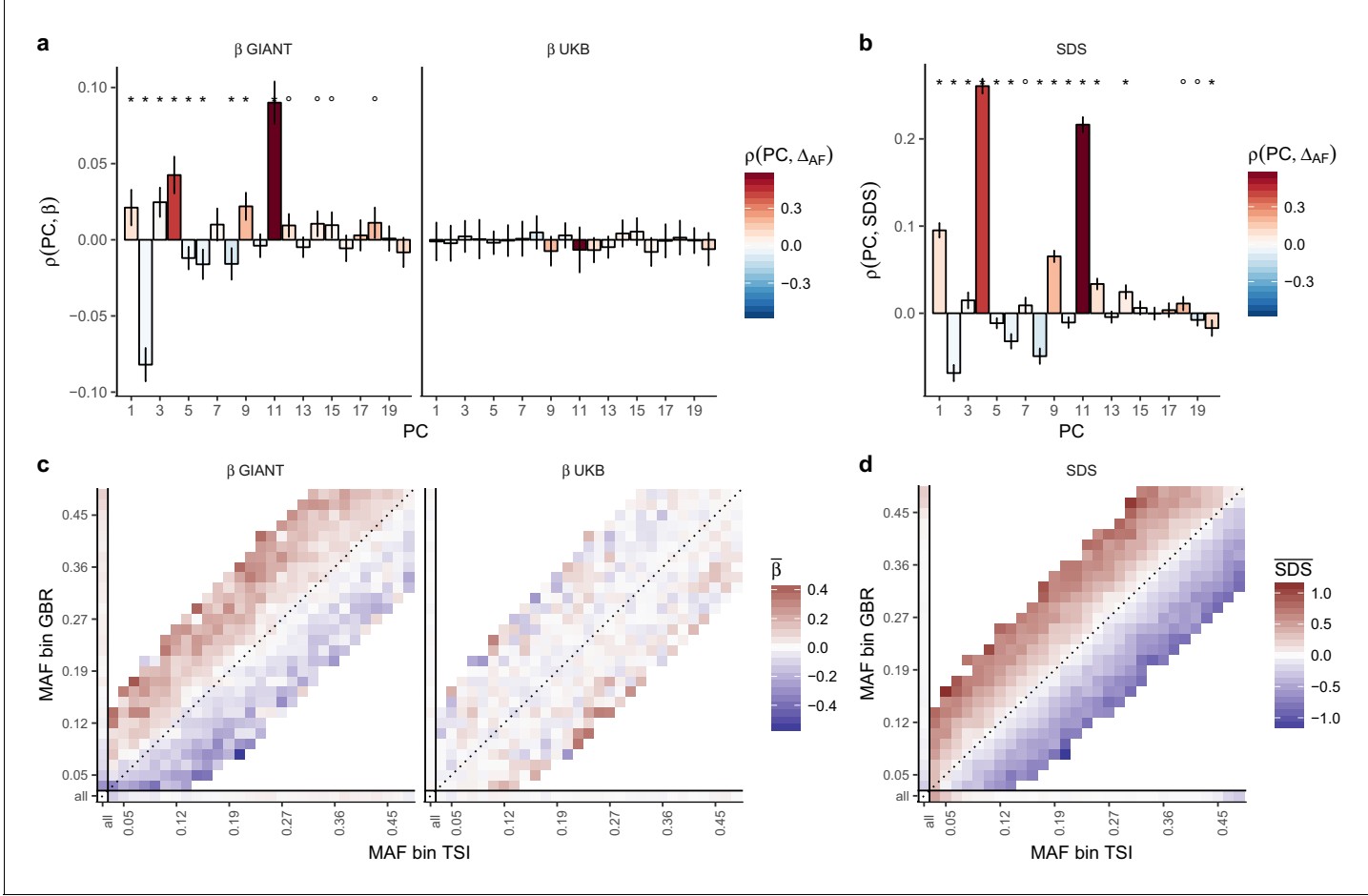

**Figure 2.** Evidence of stratification in height summary statistics. Top row: Pearson Correlation coefficients of (a) PC loadings and height beta coefficients from GIANT and UKB, and (b) PC loadings and SDS (pre-computed in the UK10K) across all SNPs. PCs were computed in all 1000 genomes phase one samples (***Abecasis et al., 2012***). Colors indicate the correlation of each PC loading with the allele frequency difference between GBR and TSI, a proxy for the European North-South genetic differentiation. PC 4 and 11 are most highly correlated with the GBR - TSI allele frequency difference. Confidence intervals and P values are based on Jackknife standard errors (1000 blocks). Open circles indicate correlations significant at alpha = 0.05, stars indicate correlations significant after Bonferroni correction in 20 PCs (p<0.0025). Bottom row: Heat map after binning all SNPs by GBR and TSI minor allele frequency of (c) mean beta coefficients from GIANT and UKB, and (d) SDS scores for all SNPs. Only bins with at least 300 SNPs are shown. While the stratification effect in SDS is not unexpected, it can lead to false conclusions when applied to summary statistics that exhibit similar stratification effects. See ***Figure 2—figure supplements 1–3*** for analyses of stratification effects in different summary statistics, and ***Supplementary file 3*** for further description of stratification effects. UKB height betas exhibit stratification effects that are weaker, and in the opposite direction of the stratification effects in GIANT (see ***Figure 2—figure supplement 4*** for a possible explanation). See ***Figure 2—source data 1*** for figure data.

DOI: https://doi.org/10.7554/eLife.39702.010

The following source data and figure supplements are available for figure 2:

**Source data 1.** Evidence of stratification in height summary statistics.
DOI: https://doi.org/10.7554/eLife.39702.015

**Figure supplement 1.** Pearson Correlation coefficients of PC loadings and height beta coefficients for different summary statistics.
DOI: https://doi.org/10.7554/eLife.39702.011

**Figure supplement 2.** Heat map of mean beta coefficients for different summary statistics.
DOI: https://doi.org/10.7554/eLife.39702.012

**Figure supplement 3.** Effect of GBR-TSI allele frequency difference on beta estimates and P values.
DOI: https://doi.org/10.7554/eLife.39702.013

**Figure supplement 4.** Height (cm) in the UKB as a function of GBR-TSI score.
DOI: https://doi.org/10.7554/eLife.39702.014

## Population structure within the UK biobank

We also note that the white British subset of the UKB data is not completely free of population stratification (as shown previously [*Haworth et al., 2019*]), although the magnitude of the potential confounding is much smaller than in the Continental European population (*Figure 2—figure supplements 1–2*). Interestingly, the north-south genetic cline in the UK tracks the height gradient in the opposite direction than in Continental Europe (*Figure 2—figure supplements 2* and *4*), and after correcting with principal components, we do not observe any evidence of residual stratification in comparison with the 1000 genomes data (*Figure 2a,c*). However, we cannot exclude the possibility of uncorrected population stratification, even in the UK Biobank, along axes not captured by the principal components of the 1000 genomes project data. For example, even for genome-wide significant SNPs (*Figure 1—figure supplement 2b*), polygenic scores for both modern and ancient individuals change when UKB summary statistics (WB ancestry controlling for 10 PCs) are used instead of GIANT. This shift, for example, for the ancient European hunter-gatherer polygenic score is troubling as different European populations are shown to have variable amounts of genetic ancestry from ancient 'hunter-gatherer' vs. 'early farmer' vs. 'steppe ancestry' populations (*Haak et al., 2015*; *Galinsky et al., 2016*), and could reflect residual stratification in the UKB GWAS not captured by the 1000 genomes PCs.

## Effects of population structure on within-population adaptation inference

We proceeded to investigate the effects of uncontrolled population stratification in GWAS discussed above on a coalescent approach such as tSDS that relies on singleton density (*Field et al., 2016a*). In principle, this approach is robust to the type of population stratification that affects the allele-frequency based tests. However, there is a north-south cline in singleton density in Europe due to lower genetic diversity in northern than in southern Europeans, leading to singleton density being lower in northern than in southern regions (*Sohail et al., 2017*). As a consequence, SDS tends to be higher (corresponding to fewer singletons) in alleles more common in GBR than in TSI (*Figure 2d*). This cline in singleton density coincidentally parallels the phenotypic cline in height and the major axis of genome-wide genetic variation. Therefore, when we perform the tSDS test using GIANT, we find a higher SDS around the inferred height-increasing alleles, which tend, due to the uncontrolled population stratification in GIANT, to be at high frequency in northern Europe (*Figure 2c*). This effect does not appear when we use UK Biobank summary statistics because of the much lower level of population stratification and more modest variation in height. We find that SDS is not only correlated with GBR-TSI allele frequency differences, but with several principal component loadings across all SNPs (*Figure 2b*), and that these SDS-PC correlations often coincide with correlations between GIANT-estimated effect sizes and PC loadings (*Figure 2a*). We further find that the tSDS signal which is observed across the whole range of P values in some GWAS summary statistics can be mimicked by replacing SDS with GBR-TSI allele frequency differences (*Figure 3a and c*, *Figure 3—figure supplements 1–4*), suggesting that the tSDS signal at non-significant SNPs may be driven in part by residual population stratification.

### A residual signal of polygenic adaptation on height?

For polygenic adaptation within a population, a small but significant tSDS signal is observed in the UK when we restrict to genome-wide significant SNPs ($p<5\times10^{-8}$). This effect persists when using UK Biobank family-based estimates (UKB sibs WB) for genome-wide significant SNPs (*Figure 3b*), and is not driven by allele frequency differences between GBR and TSI (*Figure 3d*), suggesting an attenuated signal of polygenic adaptation in the UK that is driven by a much smaller number of SNPs than previously thought. Indeed, under most genetic architectures, a tSDS signal which is driven by natural selection is not expected to lead to an almost linear increase over the whole P value range in a well-powered GWAS. Instead, we would expect to see a greater difference between highly significant SNPs and non-significant SNPs, similar to the pattern observed in the UK Biobank (*Figure 3a*).

For population-level differences in height, we assessed whether any remaining variation in height polygenic scores among populations is driven by polygenic adaptation by testing against a null model of genetic drift (*Berg and Coop, 2014*). We re-computed polygenic height scores in the POPRES dataset to increase power for this analysis as it has larger sample sizes of northern and

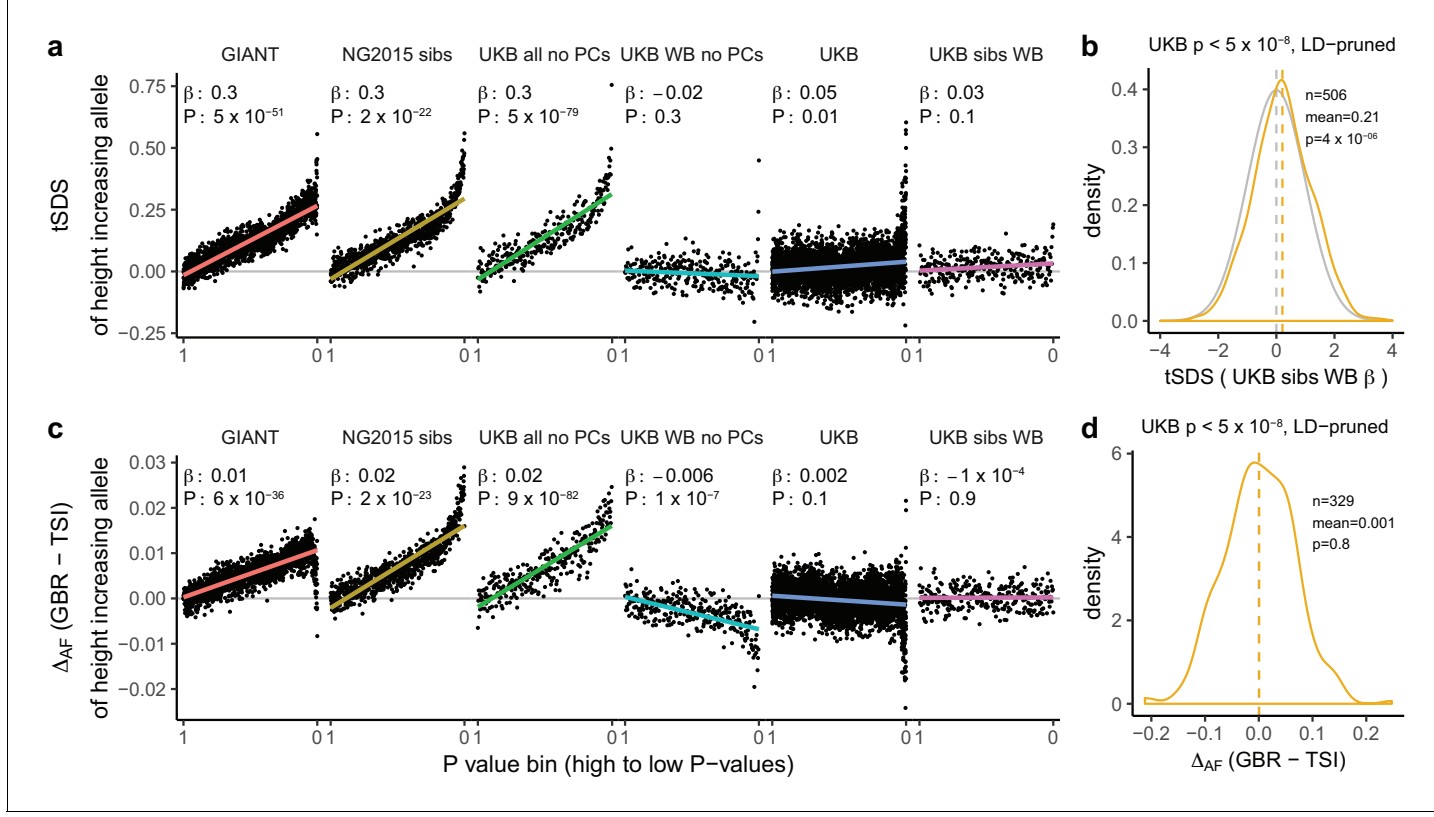

**Figure 3.** Height tSDS results for different summary statistics. (a) Mean tSDS of the height increasing allele in each P value bin for six different summary statistics. The first two panels are computed analogously to Figure 4A and Figure S22 of *Field et al. (2016a)*. In contrast to those Figures and to *Figure 1b*, the displayed betas and P values correspond to the slope and P value of the linear regression across all un-binned SNPs (rather than the Spearman correlation coefficient and Jackknife P values). The y-axis has been truncated at 0.75, and does not show the top bin for UKB all no PCs, which has a mean tSDS of 1.5. (b) tSDS distribution of the height increasing allele in 506 LD-independent SNPs which are genome-wide significant in a UKB height GWAS, where the beta coefficient is taken from a within sibling analysis in the UKB. The gray curve represents the standard normal null distribution, and we observe a significant shift. (c) Allele frequency difference between GBR and TSI of the height increasing allele in each P value bin for six different summary statistics. Betas and P values correspond to the slope and P value of the linear regression across all un-binned SNPs. The lowest P value bin in UKB all no PCs with a y-axis value of 0.06 has been omitted. (d) Allele frequency difference between GBR and TSI of the height increasing allele in 329 LD-independent SNPs which are genome-wide significant in a UKB height GWAS and were intersected with our set of 1000 genomes SNPs. There is no significant difference in frequency in these two populations, suggesting that tSDS shift at the genome-wide significant SNPs is not driven by population stratification at least due to this particular axis. The patterns shown here suggest that the positive tSDS values across the whole range of P values is a consequence of residual stratification. At the same time, the increase in tSDS at genome-wide significant, LD-independent SNPs in (b) cannot be explained by GBR - TSI allele frequency differences as shown in (d). See *Figure 3—figure supplements 1–4* for other GWAS summary statistics for unpruned and LD-pruned SNPs. Binning SNPs by P value without LD-pruning can lead to unpredictable patterns at the low P value end, as the SNPs at the low P value end are less independent of each other than higher P value SNPs (*Figure 3—figure supplement 5*). See *Figure 3—source data 1* for figure data.

DOI: https://doi.org/10.7554/eLife.39702.016

The following source data and figure supplements are available for figure 3:

**Source data 1.** Height tSDS results for different summary statistics.
DOI: https://doi.org/10.7554/eLife.39702.022
**Figure supplement 1.** tSDS for height-increasing alleles using effect sizes from different summary statistics.
DOI: https://doi.org/10.7554/eLife.39702.017
**Figure supplement 2.** Allele frequency difference for height-increasing alleles using different summary statistics.
DOI: https://doi.org/10.7554/eLife.39702.018
**Figure supplement 3.** tSDS for LD-pruned height-increasing alleles using effect sizes from different summary statistics.
DOI: https://doi.org/10.7554/eLife.39702.019
**Figure supplement 4.** Allele frequency difference for LD-pruned height-increasing alleles using different summary statistics.
DOI: https://doi.org/10.7554/eLife.39702.020
**Figure supplement 5.** Number of independent regions per GWAS P value bin in the UK Biobank.

*Figure 3 continued on next page*

*Figure 3 continued*

DOI: https://doi.org/10.7554/eLife.39702.021

southern Europeans than the 1000 Genomes project (*Nelson et al., 2008*). We computed height scores using independent SNPs that are 1) genome-wide significant in the UK Biobank ('gw-sig', $p<5\times10^{-8}$) and 2) sub-significantly associated with height ('sub-sig', $p<0.01$) in different GWAS datasets. For each of these, we tested if population differences were significant due to an overall overdispersion ($P_{Qx}$), and if they were significant along a north-south cline ($P_{lat}$) (*Figure 4*, *Figure 4—figure supplements 1–2*). Both gw-sig and sub-sig SNP-based scores computed using GIANT effect sizes showed significant overdispersion of height scores overall and along a latitude cline, consistent with previous results (*Figure 4*, *Figure 4—figure supplements 1–2*). However, the signal attenuated dramatically between sub-sig ($Q_x = 1100$, $P_{Qx} = 1\times10^{-220}$) and gw-sig ($Q_x = 48$, $P_{Qx} = 2\times10^{-4}$) height scores. In comparison, scores that were computed using the UK Biobank (UKB) effect sizes showed substantially attenuated differences using both sub-sig ($Q_x = 64$, $P_{Qx} = 5\times10^{-7}$) and gw-sig ($Q_x = 33$, $P_{Qx} = 0.02$) SNPs, and a smaller difference between the two scores. This suggests that the attenuation of the signal in GIANT is not only driven by a loss of power when using fewer gw-sig SNPs, but also reflects a decrease in stratification effects. The overdispersion signal disappeared entirely when the UK Biobank family based effect sizes were used (*Figure 4*, *Figure 4—figure supplements 1–2*). Moreover, $Q_x$ P values based on randomly ascertained SNPs and UK Biobank summary statistics are not uniformly distributed as would be expected if the theoretical null model is valid and if population structure is absent (*Figure 4—figure supplement 3*). The possibility of residual stratification effects even in the UK Biobank is also supported by a recent study (*Haworth et al., 2019*). Therefore, we remain cautious about interpreting any residual signals as 'real' signals of polygenic adaptation.

## Discussion

We have shown, by conducting a detailed analysis of human height, that estimates of population differences in polygenic scores are reduced when using the UK Biobank GWAS data relative to claims of previous studies that used GWAS meta-analyses such as GIANT. We find some evidence for population-level differences in genetic height, but it can only be robustly seen at highly significant SNPs, because any signal at less significant P values is dominated by the effect of residual population stratification. Even genome-wide significant SNPs in these analyses may be subtly affected by population structure, leading to continued overestimation of the effect. Thus, it is difficult to arrive at any quantitative conclusion regarding the proportion of the population differences that are due to statistical biases vs. population stratification of genetic height. Further, estimates of the number of independent genetic loci contributing to complex trait variation are sensitive to and likely confounded by residual population stratification.

We conclude that while effect estimates are highly concordant between GIANT and the UK Biobank when measured individually (*Supplementary file 5–7*, *Figure 1—figure supplement 1*), they are also influenced by residual population stratification that can mislead comparisons of complex traits across populations and inferences about polygenic adaptation. Although these biases are subtle, in the context of tests for polygenic adaptation, which are driven by small systematic shifts in allele frequency, they can create highly significant artificial signals especially when SNPs that are not genome-wide significant are used to estimate genetic height. Our results do not question the reliability of the genome-wide significant associations discovered in the GIANT cohort. However, we urge caution in the interpretation of signals of polygenic adaptation or between-population differences that are based on large number of sub-significant SNPs–particularly when using effect sizes derived from meta-analysis of heterogeneous cohorts which may be unable to fully control for population structure.

Our results have implications in other areas of human genetics research. For example, there is growing interest in polygenic scores that predict complex phenotypes from the aggregate effects of all allelic variants (*Wray et al., 2007*; *Purcell et al., 2009*; *Vilhjálmsson et al., 2015*; *Chun et al., 2018*). The observation that individuals with extreme values of polygenic scores exhibit many-fold elevated risk of common diseases raises hopes for their potential clinical utility (*Ganna et al., 2013*;

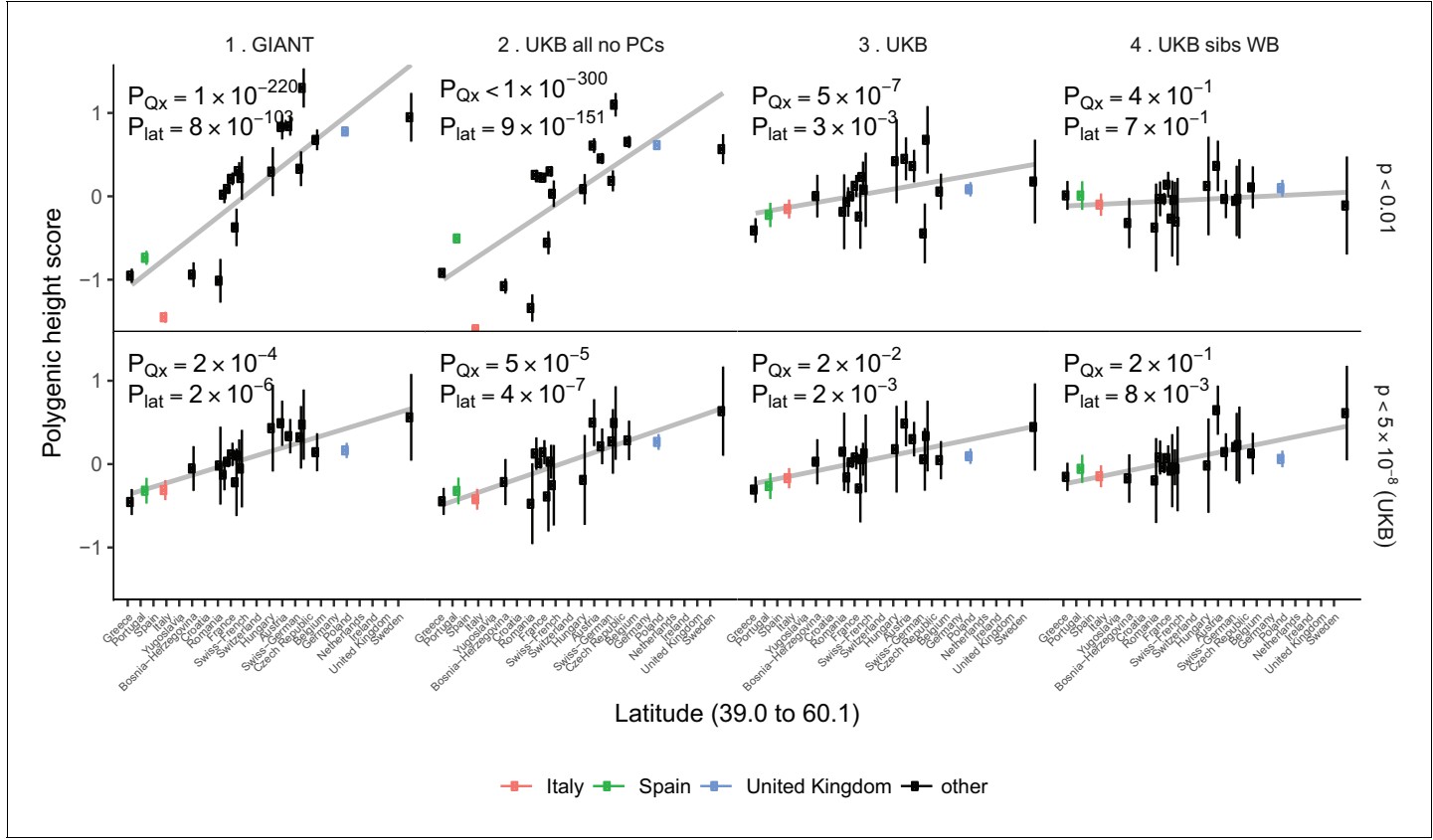

**Figure 4.** Polygenic height scores in POPRES populations show a residual albeit attenuated signal of polygenic adaptation for height. Standardized polygenic height scores from four summary statistics for 19 POPRES populations with at least 10 samples per population, ordered by latitude (see *Supplementary file 4*). The grey line is the linear regression fit to the mean polygenic scores per population. Error bars represent 95% confidence intervals and are calculated in the same way as in *Figure 1*. SNPs which were overlapping between each set of the summary statistics and the POPRES SNPs were clumped using PLINK 1.9 with parameters $r2 < 0.1$, 1 Mb distance, $p<1$. (Top) A number of independent SNPs was chosen for each summary statistic to match the number of SNPs which remained when clumping UKB at $p<0.01$. (Bottom) A set of independent SNPs with $p<5\times10^{-8}$ in the UK Biobank was selected and used to compute polygenic scores along with effect size estimates from each of the different summary statistics. The numbers on each plot show the $Q_x$ P value and the latitude covariance P value respectively for each summary statistic. See *Figure 4—figure supplements 1–4* for other clumping strategies and GWAS summary statistics. See *Figure 4—source data 1* for figure data.
DOI: https://doi.org/10.7554/eLife.39702.023

The following source data and figure supplements are available for figure 4:

**Source data 1.** Polygenic height scores in POPRES populations show a residual albeit attenuated signal of polygenic adaptation for height.
DOI: https://doi.org/10.7554/eLife.39702.028

**Figure supplement 1.** Polygenic height scores in POPRES for different summary statistics.
DOI: https://doi.org/10.7554/eLife.39702.024

**Figure supplement 2.** Test statistics for $Q_x$ (left) and latitude correlation (right) in the POPRES dataset for different summary statistics.
DOI: https://doi.org/10.7554/eLife.39702.025

**Figure supplement 3.** P value calibration in the POPRES dataset for $Q_x$ and latitude covariance tests.
DOI: https://doi.org/10.7554/eLife.39702.026

**Figure supplement 4.** Spearman correlations between polygenic height scores in the POPRES dataset computed from different summary statistics.
DOI: https://doi.org/10.7554/eLife.39702.027

*Khera et al., 2018*), and use for sociogenomics applications (*Lee et al., 2018*; *Savage et al., 2018*; *Nagel et al., 2018*). It is already clear that polygenic scores derived from European populations do not translate across populations on a global scale (*Martin et al., 2017*). Our analysis further suggests that subtle population structure, especially in GWAS that are meta-analyses of independent cohorts, could be an additional source of error in polygenic scores and affect their applicability even within

populations. We also note that other factors such as gene by environment interactions can be an alternative confounding factor for GWAS effect sizes and polygenic scores.

## Materials and methods

### Genome-wide association studies (GWAS)

We analyzed height using publicly available summary statistics that were obtained either by meta-analysis of multiple GWAS or by a GWAS performed on a single large population. We used results from the GIANT Consortium (N = 253,288) (*Wood et al., 2014*) and a GWAS performed on individuals of the UK Biobank ('UKB Neale' or simply 'UK Biobank (UKB)', N = 336,474) (*Churchhouse et al., 2017*) who derive their ancestry almost entirely from the British Isles (identified as 'white British ancestry (WB)' by the UK Biobank). The Neale lab's GWAS uses a linear model with sex and 10 principal components as covariates. We also used an independent GWAS that included all UK Biobank European samples, allowing related individuals as well as population structure ('UKB Loh', N = 459,327) (*Loh et al., 2018*). Loh *et al.*'s GWAS uses a BOLT-LMM Bayesian mixed model (*Loh et al., 2018*). Association signals from the three studies are generally correlated for SNPs that are genome-wide significant in GIANT (see *Yengo et al., 2018*).

We also used previously published family-based effect size estimates (*Robinson et al., 2015*) ('NG2015 sibs') as well as a number of test summary statistics on the UK Biobank that we generated to study the effects of population stratification. These are: 'UKB Neale new' (Similar to UKB Neale, with less stringent ancestry definition and 20 PCs calculated within sample), 'UKB all no PCs' (All UK Biobank samples included in the GWAS without correction by principal components), 'UKB all 10 PCs' (All UK Biobank samples included in the GWAS with correction by 10 principal components), 'UK WB no PCs' (Only 'white British ancestry' samples included in the GWAS without correction by principal components), 'UKB WB 10 PCs' (Only 'white British ancestry' samples included in the GWAS with correction by 10 principal components), 'UKB sibs all' (All UK Biobank siblings included in the GWAS), 'UKB sibs WB' (Only UK Biobank 'white British ancestry' siblings included in the GWAS) (Please see *Supplementary file 1* for sample sizes and other details).

### Population genetic data for ancient and modern samples

We analyzed ancient and modern populations for which genotype data are publicly available. For ancient samples (*Haak et al., 2015*; *Mathieson et al., 2018*), we computed scores after dividing populations into three previously described broad ancestry labels (HG = Hunter Gatherer (n = 162 individuals), EF = Early Farmer (n = 485 individuals), and SP = Steppe Ancestry (n = 465 individuals)). For modern samples available through the 1000 genomes phase three release (*Auton et al., 2015*), we computed scores in two populations each from Northern Europe (GBR, CEU), Southern Europe (IBS, TSI), Africa (YRI, LWK), South Asia (PJL, BEB) and East Asia (CHB, JPT) (*Figure 1a*). In total, we analyzed 1112 ancient individuals, and 1005 modern individuals from 10 different populations in the 1000 genomes project (*Supplementary file 2*). We used the allele frequency differences between the GBR and TSI populations for a number of analyses to study population stratification (*Figures 2–3*). We also analyzed 19 European populations from the POPRES (*Nelson et al., 2008*) dataset with at least 10 samples per population (*Figure 4—figure supplement 4*).

All ancient samples had 'pseudo-haploid' genotype calls at 1240k sites generated by selecting a single sequence randomly for each individual at each SNP (*Mathieson et al., 2018*). Thus, there is only a single allele from each individual at each site, but adjacent alleles might come from either of the two haplotypes of the individual. We also re-computed scores in present-day 1000 genomes individuals using only pseudo-haploid calls at 1240 k sites to allow for a fair comparison between ancient and modern samples (*Figure 1—figure supplement 6*).

### Polygenic scores

The polygenic scores, confidence intervals and test statistics (against the null model of genetic drift) were computed based on the methodology developed in references *Berg and Coop, 2014* and *Berg et al., 2017*. We computed the polygenic score (Z) for a trait in a population by taking the sum of allele frequencies in that population across all L sites associated with the trait, weighting each allele's frequency ($p_l$) by its effect on the trait ($\beta_l$).

$$Z = \sum_{l}^{L} \beta_l p_l$$

Al polygenic scores are plotted in centered standardized form ($\frac{Z-\mu}{\sqrt{V_A}}$),

where $\mu = \sum_l \beta_l \bar{p}_l$, $V_A = \sum_l \beta_l^2 \bar{p}_l (1 - \bar{p}_l)$, and $\bar{p}_l$ is the mean allele frequency across all populations analyzed. Source code repositories for the polygenic score analysis and computing scripts and source data for all the main figures have been made available at https://github.com/msohail88/poly-genic_selection (*Sohail, 2018*; copy archived at https://github.com/elifesciences-publications/poly-genic_selection) and https://github.com/uqrmaie1/sohail_maier_2019 (*Sohail, 2019*; copy archived at https://github.com/elifesciences-publications/sohail_maier_2019).

Polygenic scores were computed using independent GWAS SNPs associated with height in three main ways: (1) The genome was divided into ~1700 non-overlapping linkage disequilibrium (LD) blocks (using the approximately independent linkage disequilibrium blocks in the EUR population computed in *Berisa and Pickrell, 2015*), and the SNP with the lowest P value within each block was picked to give a set of ~1700 independent SNPs for each height GWAS used (all SNPs for which effect sizes are available were considered) similar to the analysis in *Berg et al., 2017*. In (2) and (3), Plink's (*Chang et al., 2015*; *Purcell and Chang, 2015*) clumping procedure was used to make independent 'clumps' of SNPs for each GWAS at different P value thresholds. This procedure selects SNPs below a given P value threshold as index SNPs to start clumps around, and then reduces all SNPs below a given P value threshold that are in LD with these index SNPs (above an $r^2$ threshold, 0.1) and within a physical distance of them (1 Mb) into clumps with them. Clumps are preferentially formed around index SNPs with the lowest P value in a greedy manner. The index SNP from each clump is then picked for further polygenic score analyses. The algorithm is also greedy such that each SNP will only appear in one clump if at all. We clumped each GWAS to obtain (2) a set of independent sub-significant SNPs associated with height (p<0.01) similarly to *Robinson et al. (2015)*, and (3) a set of independent genome-wide significant SNPs associated with height (p<$5\times10^{-8}$). The 1000 genomes phase three dataset was used as the reference panel for computing LD for the clumping procedure.

The estimated effect sizes for these three sets of SNPs from each GWAS was used to compute scores. Only autosomal SNPs were used for all analyses to avoid creating artificial mean differences between populations with different numbers of males and females.

The 95% credible intervals were constructed by assuming that the posterior of the underlying population allele frequency is independent across loci and populations and follows a beta distribution. We updated a Uniform prior distribution with allele counts from ancient and modern populations to obtain the posterior distribution at each locus in each population. We estimated the variance of the polygenic score $V_Z$ using the variance of the posterior distribution at each locus, and computed the width of 95% credible intervals as $1.96\sqrt{V_Z}$ for each population.

The $Q_x$ test statistic measures the degree of overdispersion of the mean population polygenic score compared to a null model of genetic drift. It assumes that the vector of mean centered mean population polygenic score follows a multivariate normal distribution: Z ~ MVN(0, 2 $V_A$ F), where $V_A$ is the additive genetic variance of the ancestral population and F is a square matrix describing the population structure. This is equivalent to the univariate case of the test statistic used in *Robinson et al. (2015)*. The latitude test statistic assumes that Y'Z ~ N(0, 2 $V_A$ Y'FY), where Y is a mean centered vector of latitudes for each population (*Berg et al., 2019*).

## tSDS analysis

The Singleton Density Score (SDS) method identifies signatures of recent positive selection based on a maximum likelihood estimate of the log-ratio of the mean tip-branch length of the derived vs. the ancestral allele at a given SNP. The tip-branch lengths are inferred from the average distance of each allele to the nearest singleton SNP across all individuals in a sequencing panel. When the sign of the SDS scores is aligned with the trait-increasing or trait-decreasing allele in the effect estimates of a GWAS, the Spearman correlation between the resulting tSDS scores and the GWAS P values has been proposed as an estimate of recent positive selection on polygenic traits.

Here, we applied the tSDS method using pre-computed Singleton Density Scores for 4,451,435 autosomal SNPs obtained from 3195 individuals from the UK10K project (*Field et al., 2016a*; *Field et al., 2016b*) for SNPs associated with height in GIANT and the UK biobank (*Figure 1b*) and in different summary statistics (*Figure 3*). After normalizing SDS scores in each 1% allele frequency bin to mean zero and unit variance, excluding SNPs from the MHC region on chromosome six and aligning the sign of the SDS scores to the height increasing alleles (resulting in tSDS scores), we computed the Spearman correlation coefficient between the tSDS score and the GWAS P value. The tSDS Spearman correlation standard errors and P values were computed using a block-jackknife approach, where each block of 1% of all SNPs ordered by genomic location was left out and the Spearman correlation coefficient was computed on the remaining SNPs. We also compared the tSDS score distributions for only genome-wide significant SNPs (*Figure 3b*).

## Population structure analysis

To compute SNP loadings of the principal components of population structure (PC loadings) in the 1000 genomes data (*Figure 2*), we first computed PC scores for each individual. We used SNPs that had matching alleles in 1000 genomes, GIANT and UK Biobank, that had minor allele frequency >5% in 1000 genomes, and that were not located in the MHC locus, the chromosome eight inversion region, or regions of long LD. After LD pruning to SNPs with $r^2$ <0.2 relative to each other, PCA was performed in PLINK on the 187,160 remaining SNPs. In order to get SNP PC loadings for more SNPs than those that were used to compute PC scores, we performed linear regressions of the PC scores on the genotype allele count of each SNP (after controlling for sex) and used the resulting regression coefficients as the SNP PC loading estimates. The 1000 genomes phase one dataset (*Abecasis et al., 2012*) was used to compute the PC loadings.

## Acknowledgements

We thank Alkes Price, Jeremy Berg, Graham Coop, Jonathan Pritchard, Matthew Robinson, Jian Yang, Peter Visscher, Hilary Finucane, John Novembre and Raymond Walters for useful discussions and comments that significantly improved the manuscript. The study was supported by National Institutes of Health grants HG009088, MH101244 (MS, RM, BN and SS) and GM127131 (SS). DR was supported by National Institutes of Health grants GM100233 and HG006399, an Allen Discovery Center grant from the Paul Allen Foundation, and the Howard Hughes Medical Institute. IM was supported by a Sloan Research Fellowship and a New Investigator Research Grant from the Charles E Kaufman foundation.

This research was conducted using the UK Biobank Resource applications 18597, 11898 and 31063.

## Additional information

### Competing interests

Benjamin Neale: Ben Neale is a member and on the scientific advisory board of Deep Genomics, a consultant for Camp4 Therapeutics Corporation, a consultant for Merck & Co., a consultant for Takeda Phamaceutical, and a consultant for Avanir Pharmaceuticals. None of these entities played a role in determining the content of this paper. The other authors declare that no competing interests exist.

### Funding

| Funder | Grant reference number | Author |
| --- | --- | --- |
| National Institutes of Health | HG009088 | Mashaal Sohail<br>Robert M Maier<br>Benjamin Neale<br>Shamil R Sunyaev |

| National Institutes of Health | MH101244 | Mashaal Sohail<br>Robert M Maier<br>Benjamin Neale<br>Shamil R Sunyaev |
| --- | --- | --- |
| Alfred P. Sloan Foundation | Sloan Research Fellowship | Iain Mathieson |
| Charles E Kaufman Foundation | New Investigator Research Grant | Iain Mathieson |
| Paul Allen Foundation | Allen Discovery Center | David Reich |
| National Institutes of Health | GM100233 | David Reich |
| National Institutes of Health | HG006399 | David Reich |
| Howard Hughes Medical Institute | Investigator | David Reich |
| National Institutes of Health | GM127131 | Shamil R Sunyaev |

The funders had no role in study design, data collection and interpretation, or the decision to submit the work for publication.

### Author contributions

Mashaal Sohail, Robert M Maier, Conceptualization, Formal analysis, Investigation, Visualization, Methodology, Writing—original draft, Writing—review and editing; Andrea Ganna, Formal analysis, Writing—review and editing; Alex Bloemendal, Mark J Daly, Nick Patterson, Methodology, Writing—review and editing; Alicia R Martin, Data curation, Writing—review and editing; Michael C Turchin, Charleston WK Chiang, Validation, Writing—review and editing; Joel Hirschhorn, Validation, Methodology, Writing—review and editing; Benjamin Neale, Supervision, Visualization, Methodology, Writing—review and editing; Iain Mathieson, Data curation, Supervision, Investigation, Methodology, Writing—review and editing; David Reich, Conceptualization, Supervision, Visualization, Methodology, Writing—original draft, Project administration, Writing—review and editing; Shamil R Sunyaev, Conceptualization, Supervision, Methodology, Writing—original draft, Project administration, Writing—review and editing

### Author ORCIDs

Mashaal Sohail (iD) https://orcid.org/0000-0002-6586-4403
Robert M Maier (iD) http://orcid.org/0000-0002-3044-090X
Michael C Turchin (iD) http://orcid.org/0000-0003-3569-1529
Charleston WK Chiang (iD) https://orcid.org/0000-0002-0668-7865
Shamil R Sunyaev (iD) https://orcid.org/0000-0001-5715-5677

### Decision letter and Author response

Decision letter https://doi.org/10.7554/eLife.39702.041
Author response https://doi.org/10.7554/eLife.39702.042

## Additional files

### Supplementary files

• Supplementary file 1. Description of 11 GWAS summary statistics.
DOI: https://doi.org/10.7554/eLife.39702.029

• Supplementary file 2. Table of ancient and 1000 genomes modern populations used with sample sizes.
DOI: https://doi.org/10.7554/eLife.39702.030

• Supplementary file 3. Supplementary note on characterization of stratification effects in GIANT and UK Biobank.
DOI: https://doi.org/10.7554/eLife.39702.031

• Supplementary file 4. Table of POPRES populations used with sample sizes and latitude.
DOI: https://doi.org/10.7554/eLife.39702.032

• Supplementary file 5. LD Score regression estimates for 11 different summary statistics. LD score regression can be used to detect residual stratification effects in summary statistics, and so we tested whether LDSC confirms our hypothesis of residual stratification. We detect a greatly inflated intercept estimate of 9.42 in UKB all no PCs, but only a moderately increased intercept value in GIANT and an intercept less than one in NG2015 sibs. The relatively small GIANT intercept can be explained by cohort-wise lambda-GC correction, while the low intercept in NG2015 sibs is possibly caused by the adaptive permutation procedure which does not compute precise p-values for non-significant associations. In both cases LDSC cannot be expected to pick up stratification effects, since the generation of summary statistics is not in line with the LDSC model.
DOI: https://doi.org/10.7554/eLife.39702.033

• Supplementary file 6. Correlation of beta estimates at all 86,153 shared SNPs.
DOI: https://doi.org/10.7554/eLife.39702.034

• Supplementary file 7. Correlation of beta estimates at 2251 shared SNPs which are significant in the UK Biobank.
DOI: https://doi.org/10.7554/eLife.39702.035

• Transparent reporting form
DOI: https://doi.org/10.7554/eLife.39702.036

### Data availability

All newly generated UK Biobank height GWAS summary statistics have been made available at http://dx.doi.org/10.5061/dryad.8g5g6j4. Results from the GIANT Consortium (GWAS Anthropometric 2014 Height) were downloaded from https://portals.broadinstitute.org/collaboration/giant/index.php/GIANT_consortium_data_files#GWAS_Anthropometric_2014_Height. GWAS results from the UK Biobank ("UKB" or "UKB Neale") were downloaded from http://www.nealelab.is/uk-biobank. The previously published family-based effect size estimates ("NG2015 sibs") can be accessed here http://cnsgenomics.com/data/robinson_et_al_2015_ng/withinfam_summary_ht_bmi_release_March2016.tar.gz. The independent mixed model association analysis that included all UK Biobank individuals of European ancestry ("UKB Loh") was downloaded from https://data.broadinstitute.org/alkesgroup/UKBB/body_HEIGHTz.sumstats.gz. Approximately independent linkage disequilibrium blocks in human populations were downloaded for the EUR population from https://bitbucket.org/nygcresearch/ldetect-data/overview. Source code repositories for the polygenic score analysis in this manuscript and computing scripts and source data for all the main figures have been made available at https://github.com/msohail88/polygenic_selection and https://github.com/uqrmaie1/sohail_maier_2019 (copies archived at https://github.com/elifesciences-publications/polygenic_selection and https://github.com/elifesciences-publications/sohail_maier_2019, respectively).

The following dataset was generated:

| Author(s) | Year | Dataset title | Dataset URL | Database and Identifier |
|---|---|---|---|---|
| Sohail M, Maier RM, Ganna A | 2018 | Data from: Polygenic adaptation on height is overestimated due to uncorrected stratification in genome-wide association studies | http://dx.doi.org/10.5061/dryad.8g5g6j4 | Dryad Digital Repository, 10.5061/dryad.8g5g6j4 |

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
