## [Decision Letter]

[**Editorial note:** This article has been through an editorial process in which the authors decide how to respond to the issues raised during peer review. The Reviewing Editor's assessment is that all the issues have been addressed.]

Thank you for submitting your article "Signals of polygenic adaptation on height have been overestimated due to uncorrected population structure in genome-wide association studies" for consideration by *eLife*. Your article has been reviewed by three peer reviewers, including Magnus Nordberg as the Reviewing Editor and Reviewer #1, and the evaluation has been overseen by Mark McCarthy as the Senior Editor. The following individuals involved in review of your submission have also agreed to reveal their identity: Nicholas H Barton (Reviewer #2); Joachim Hermisson (Reviewer #3).

The Reviewing Editor has summarized the major concerns shared by all reviewers, and we have also included the separate reviews below for your consideration.

If you have any questions, please do not hesitate to contact us.

Summary:

This is one of two papers demonstrating that published signals of selection on human height cannot be replicated in the recently released UK Biobank data, apparently because these signals were caused by confounding population structure that is absent in UK Biobank data.

Major concerns:

We were struck by how both papers focus on spurious signals of selection rather than the underlying cause, which is that the GWAS effect-size estimates are confounded. The former is a somewhat esoteric question, but the latter may have enormous implications for much of human genetics, and these papers are likely to be heavily cited because of this. However, the papers seem to go out of their way to avoid discussing this topic. Of course we are not the authors, but, for the record, it looks odd.

Furthermore, the papers seem to suggest that confounding is not present in the UK Biobank data, but isn't it more likely that the magnitude is simply smaller?

Both papers also present evidence that a sib-based study by Robinson et al., 2015, that was meant to eliminate confounding did no such thing. This is disturbing, and while we understand that identifying the reason may be beyond the present papers, the general implications should again probably be discussed.

Finally, this paper often seems stream-of-consciousness: it lacks detailed explanations as well as a coherent outline, making it very difficult to follow unless you are a specialist in the field. We urge the authors to explain better for a general audience.

Separate reviews (please respond to each point):

*Reviewer #1:*

This one of at least two papers appearing simultaneously and reaching exactly the same conclusion. It is well written.

The only thing that surprises me about this paper is that it, as well as the other one I have seen, focuses on the relatively obscure issue of whether height has been under selection, tiptoeing around the much bigger issue (the elephant in the room) that the reason the claims for selection do not stand is that the GWAS estimates of effect sizes are biased because of population structure. It is not just the selection signals that do not replicate, but the polygenic scores. I'm not surprised, but, as you know, there are probably at least a hundred papers out there that are based on the infallibility of LD score regression and genomic prediction. I understand the need for caution before attacking this edifice, but I nonetheless think some clarification is unavoidable.

*Reviewer #2:*

This paper identifies a discrepancy between signs of selection estimated from UK Biobank data, compared with previous studies, and suggest that those earlier signals were caused by subtle stratification in the data. This is a useful contribution to an important question. I have only minor comments (below), but overall, urge the authors to try to rewrite the text to make it more accessible to those not immersed in the field. I find it hard to make specific suggestions, but it comes across as a list of statistical tests, without enough flow to carry the reader along with the argument. Admittedly, given the quite intricate arguments, this is not easy to do.

Minor Comments:

Why should one believe that using the first 15 PC corrects for stratification? Even this is somehow traditional in the field, it needs explanation, since the failure of the correction is the key point of the paper.

Figure 1B: – The x axis needs to be labelled. More important, Spearman's correlation seems far too small, given that by eye, the points follow the linear regression rather well. This may be related to the large values seen at the right of each figure, fitting a single regression is clearly inappropriate. There needs to be a test which separates these two sets of points in some way: as it stands the significance test is just not appropriate.

Figure 3: – b in the figure should be β. Also, there is a paragraph break before "The patterns" which makes it hard to work out what is main text and what is caption.

Figure 4: – I do not understand what the "six summary statistics" are here.

Discussion section: – The concluding paragraph seems too weak, especially the sentence "In no way..". Surely the point of the paper is to "question the statistical methodology.… in polygenic tyests for adaptation", since that methodology seems to give spurious results? It is also not at all clear how much the stratification implied here influences effect size estimates in GWAS.

Paragraph five of subsection “Polygenic scores”: Does the distribution in fact follow a β?

*Reviewer #3:*

Both manuscripts by Berg et al. and Sohail et al. present thorough and insightful analyses with highly relevant results for current and future GWAS studies. Even prior to publication, the manuscripts have considerable impact. They will be widely read and cited. I do not think that further analyses are needed, with the potential exception of the third point below. All other points concern the discussion, in particular the guidance for further research that will surely emerge from these studies.

How safe are results based on the UK Biobank data?

This refers to the weak signals reported (with much caution) in the present studies, but also to potential future results on other traits. You recommend using data "such as UKB" and we will certainly see many more studies based on this resource. I would therefore appreciate a more specific discussion of risks connected to this particular data set.

1) Stratification even within the UKB-GB data: It is well known that height and socioeconomic status are correlated in modern societies (e.g. BMJ 2016; 352:i582), and social status correlates with descent. In the UK, both factors are also geographically stratified, with people living in the north of the country having lower socioeconomic status and shorter stature, on average, than those in the south. Furthermore, the percentage of Anglo-Saxon admixture varies across the UK. How could these factors influence results based on UKB data, both here and otherwise?

2) Potential influence of GxE interactions: The manuscripts focus (for good reason) on issues connected with stratification. However, if polygenic scores depend on the environment (e.g., due to countergradient variation), GxE interactions are an alternative confounding factor. Importantly, use of a homogeneous detection panel (to avoid stratification), such as UKB-GB, could increase these effects. Maybe this should be briefly discussed in the context of the present results and mentioned as a necessary caveat also for future studies that use detection panels from narrow geographic regions.

What, exactly, causes the problems with the previous data?

3) There seem to be two relevant differences of the GIANT data relative to UKB: 1) UKB is much more homogeneous and 2) GIANT is a meta-study, collecting summary statistics from many sources that are individually corrected for stratification. One would like to know better which factor is decisive. This could be further addressed by combining summaries from sub-samples of the "UKB-all" data in an artificial meta-study.

4) The Robinson et al., 2015 GWAS: Sib-based studies are done to avoid / minimize stratification effects and the Robinson 2015 data have been used as a proof of robustness in several previous studies. The fact that you find clear signs of stratification is sobering and one would like to know what has gone wrong. You may not currently have any explanation and this is fair enough. However, the discussion should be clearer and say upfront that results based on these data cannot be trusted until we understand the issues.

Minor Comments:

a) You use 11 different summary statistics, with partly inconsistent naming strategy. I had to look up names in the methods part a number of times. I think this can be improved. Maybe even use the same names as Berg et al. where the summaries are identical.

b) The switch from 1000 genomes to POPRES complicates comparison between figures. If there are advantages of POPRES, why not use it throughout? This holds, in particular, for the test of the latitudinal slope, which would be more convincing with many populations rather than just 4 from the 1000 genomes data.

c) Figure 4: "The overdispersion signal disappeared entirely when the UK Biobank family based effect sizes were used": Is this due to the smaller sample size of the sib data or due to residual stratification issues in UKB? This could be tested using a sub-sample from UKB of the same size as the sib data.

d) Figure 3 legend: "suggesting that tSDS shift at the gw-significant SNPs is not driven by population stratification": only true for stratification due to this particular axis.

Additional data files and statistical comments:

All necessary information is provided and the UKB sib data is on Dryad. I think the other newly generated GWAS data should go there, too.

---

## [Author Response]

Major concerns:We were struck by how both papers focus on spurious signals of selection rather than the underlying cause, which is that the GWAS effect-size estimates are confounded. The former is a somewhat esoteric question, but the latter may have enormous implications for much of human genetics, and these papers are likely to be heavily cited because of this. However, the papers seem to go out of their way to avoid discussing this topic. Of course we are not the authors, but, for the record, it looks odd.

We agree that our analysis raises broader issues beyond detection of polygenic adaptation. However, we do not find that our results question the whole GWAS enterprise. Almost all genome-wide significant signals identified by the GIANT consortium replicate in the UK Biobank. Overall, there is a high correlation between the effect size estimates between the two studies. However, certain aspects of current human genetics research outside evolutionary biology are obviously affected. The prime example is the transferability of polygenic risk scores between populations. We added a detailed discussion of this in the revised manuscript.

Furthermore, the papers seem to suggest that confounding is not present in the UK Biobank data, but isn't it more likely that the magnitude is simply smaller?

The revised version of the manuscript discusses the confounding in the UK biobank data. We clearly demonstrate that uncorrected summary statistics of the UK Biobank GWAS show signals of stratification even if the analysis is restricted to White British individuals. Interestingly, in the UK the north-south genetic cline tracks the height gradient in the opposite direction than in Continental Europe. Obviously, the magnitude of the confounding is much smaller. When principal components are included in the UK Biobank GWAS, we do not find any evidence of residual stratification when testing for a correlation between effect size estimates and twenty 1000 genomes principal components (Figure 2). However, this does not preclude the possibility of residual stratification along axes that are not captured by these principal components.

Both papers also present evidence that a sib-based study by Robinson et al., 2015, that was meant to eliminate confounding did no such thing. This is disturbing, and while we understand that identifying the reason may be beyond the present papers, the general implications should again probably be discussed.

In the revised manuscript, we clarify that we agree with the conceptual approach of Robinson et al. but that the discrepancy is likely to be due to a technical error in the computations of Robinson et al. We have in fact now confirmed this through correspondence with the authors of Robinson et al., and they are currently preparing a manuscript revisiting these analyses and correcting the technical issues. We emphasize that family-based effect size estimates computed in the UK Biobank following the Robinson et al. methodology behave as expected.

Finally, this paper often seems stream-of-consciousness: it lacks detailed explanations as well as a coherent outline, making it very difficult to follow unless you are a specialist in the field. We urge the authors to explain better for a general audience.

We edited the text making it more accessible.

Separate reviews (please respond to each point):

Reviewer #1:

This one of at least two papers appearing simultaneously and reaching exactly the same conclusion. It is well written.The only thing that surprises me about this paper is that it, as well as the other one I have seen, focuses on the relatively obscure issue of whether height has been under selection, tiptoeing around the much bigger issue (the elephant in the room) that the reason the claims for selection do not stand is that the GWAS estimates of effect sizes are biased because of population structure. It is not just the selection signals that do not replicate, but the polygenic scores. I'm not surprised, but, as you know, there are probably at least a hundred papers out there that are based on the infallibility of LD score regression and genomic prediction. I understand the need for caution before attacking this edifice, but I nonetheless think some clarification is unavoidable.

As noted above, we added a discussion on the potential impact of our findings on the debate about the transferability of polygenic scores between populations. In this work we have focused on the effects of residual population stratification on tests of selection because they appear to be particularly sensitive. While we cannot exclude the possibility that some other methods are also sensitive to residual stratification, neither our analyses nor previous publications provide evidence that this is a widespread problem for other applications of GWAS data, even though our results certainly do highlight the importance of revisiting the importance of population stratification in all analyses of polygenic predictors. As for polygenic scores, it has been demonstrated previously that results can be unreliable when predicting across populations (Martin et al., 2017). The fact that polygenic scores from the UK Biobank tend to have a higher out of sample prediction accuracy than polygenic scores from GIANT is just one of many pieces of evidence showing that polygenic scores do not just pick up residual stratification, but rather a signal from the trait of interest.

Generally, any method that uses genetic data from multiple populations is prone to be susceptible to bias from residual stratification. However, our SDS results show that even methods which do not use data from distinct populations can be affected by residual stratification. In the case of SDS, the problem is that both the singleton density scores and GIANT height summary statistics are stratified across the European north south cline. We cannot exclude the possibility that similar biases can exist in other methods, but we have not found any other examples of it. LD score regression is unlikely to be affected, since residual (environmental) stratification should affect high and low LD score SNPs to a similar extent, and thus not have a large impact on the parameter estimates. To confirm the robustness of LD score regression, we have compared bivariate LD score regression estimates from GIANT to estimates from the UK Biobank. In each case, we used LD hub to obtain genetic correlation estimates between height and 832 other traits. We found very high concordance between the estimates in GIANT and UK Biobank, which further supports that bivariate LD scores regression results are in fact robust to population stratification.

While it is outside the scope of this work to provide a complete characterization of the extent to which residual population stratification affects the plethora of methods that make use of GWAS summary data, we hope that it provides a useful case study and stimulates more research in this area, as well as more careful study design in general.

Reviewer #2:

This paper identifies a discrepancy between signs of selection estimated from UK Biobank data, compared with previous studies, and suggest that those earlier signals were caused by subtle stratification in the data. This is a useful contribution to an important question. I have only minor comments (below), but overall, urge the authors to try to rewrite the text to make it more accessible to those not immersed in the field. I find it hard to make specific suggestions, but it comes across as a list of statistical tests, without enough flow to carry the reader along with the argument. Admittedly, given the quite intricate arguments, this is not easy to do.

We have substantially edited the text and hope that it has become more accessible.

Minor Comments:Why should one believe that using the first 15 PC corrects for stratification? Even this is somehow traditional in the field, it needs explanation, since the failure of the correction is the key point of the paper.

One can certainly envision a scenario where very few samples in the dataset have a substantially different ancestry or there is an axis of very mild stratification affecting many samples. Both of these would not be captured by top principal components. However, investigation of these effects is not a focus of this manuscript, which attracts attention to a problem rather than suggests new analytical standards.

Figure 1B: The x axis needs to be labelled. More important, Spearman's correlation seems far too small, given that by eye, the points follow the linear regression rather well. This may be related to the large values seen at the right of each figure, fitting a single regression is clearly inappropriate. There needs to be a test which separates these two sets of points in some way: as it stands the significance test is just not appropriate.

We have labeled the x-axis for Figure 1B. We agree with the reviewer and, in our work, present an analysis that studies the tSDS distribution for only genome-wide SNPs (Figure 3). We use the Spearman correlation coefficient to recapitulate the original analysis. The points and the linear slope in the figure are for visualization only. The Spearman correlation coefficient is computed based on raw data and is unrelated to the dots that correspond to binned data.

Figure 3: b in the figure should be β. Also, there is a paragraph break before "The patterns" which makes it hard to work out what is main text and what is caption.

We have updated the figure and removed the paragraph break.

Figure 4: I do not understand what the "six summary statistics" are here.

We have corrected the caption to say “four” summary statistics

Discussion section: The concluding paragraph seems too weak, especially the sentence "In no way..". Surely the point of the paper is to "question the statistical methodology.… in polygenic tyests for adaptation", since that methodology seems to give spurious results? It is also not at all clear how much the stratification implied here influences effect size estimates in GWAS.

We have edited this sentence as well as the concluding paragraphs in general.

Paragraph five of subsection “Polygenic scores” Does the distribution in fact follow a β?

Again, we have re-implemented the existing approach used by previous work (Berg et al., 2017). Although not shown in the paper, we have investigated the issue and found that different ways to compute confidence intervals yield very similar results (including methods not making specific assumptions about the shape of allele frequency distribution).

Reviewer #3:

Both manuscripts by Berg et al. and Sohail et al. present thorough and insightful analyses with highly relevant results for current and future GWAS studies. Even prior to publication, the manuscripts have considerable impact. They will be widely read and cited. I do not think that further analyses are needed, with the potential exception of the third point below. All other points concern the discussion, in particular the guidance for further research that will surely emerge from these studies.How safe are results based on the UK Biobank data?This refers to the weak signals reported (with much caution) in the present studies, but also to potential future results on other traits. You recommend using data "such as UKB" and we will certainly see many more studies based on this resource. I would therefore appreciate a more specific discussion of risks connected to this particular data set.1) Stratification even within the UKB-GB data: It is well known that height and socioeconomic status are correlated in modern societies (e.g. BMJ 2016; 352:i582), and social status correlates with descent. In the UK, both factors are also geographically stratified, with people living in the north of the country having lower socioeconomic status and shorter stature, on average, than those in the south. Furthermore, the percentage of Anglo-Saxon admixture varies across the UK. How could these factors influence results based on UKB data, both here and otherwise?

This is an important point, which we took into account. We repeat here the same response as above (the Major comments section). The revised version of the manuscript discusses the confounding in the UK biobank data. We clearly demonstrate that uncorrected summary statistics of the UK Biobank GWAS show signals of stratification even if the analysis is restricted to White British individuals. Interestingly, in the UK the north-south genetic cline tracks the height gradient in the opposite direction than in Continental Europe. Obviously, the magnitude of the confounding is much smaller, compared to the confounding that can arise from residual stratification in a trans-European sample. It seems likely that both genetic and environmental stratification are present along similar dimensions in the UK Biobank population, which could explain this finding. Principal components can correct for stratification effects, regardless of whether they are of genetic or environmental origin. We therefore have made no attempts to disentangle the causes of stratification in the UK Biobank, but other researchers have (Haworth et al.).

2) Potential influence of GxE interactions: The manuscripts focus (for good reason) on issues connected with stratification. However, if polygenic scores depend on the environment (e.g., due to countergradient variation), GxE interactions are an alternative confounding factor. Importantly, use of a homogeneous detection panel (to avoid stratification), such as UKB-GB, could increase these effects. Maybe this should be briefly discussed in the context of the present results and mentioned as a necessary caveat also for future studies that use detection panels from narrow geographic regions.

We thank the reviewer for bringing this up and agree that GxE is another potential confounder affecting transferability of polygenic scores. At the same time, presence of GxE interactions without selection are not expected to generate a non-zero covariance between effect size estimates and allele frequency differences (basis of Qx) or between effect size estimates and allelic ages (basis of tSDS), although such interactions are an important caveat for how adaptation signals should be interpreted. We think that the countergradient due to stabilizing selection in a changing environment (with or without presence of GxE) can indeed lead to a signal of adaptation. Whether this should be considered a false signal or as a true adaptation to the same phenotypic value might be a matter of a terminology debate.

We briefly mention the potential effect of GxE on the transferability of polygenic scores in the Discussion. We decided not to include the technical points above in the manuscript because we are trying to make it more accessible.

What, exactly, causes the problems with the previous data?3) There seem to be two relevant differences of the GIANT data relative to UKB: 1) UKB is much more homogeneous and 2) GIANT is a meta-study, collecting summary statistics from many sources that are individually corrected for stratification. One would like to know better which factor is decisive. This could be further addressed by combining summaries from sub-samples of the "UKB-all" data in an artificial meta-study.

We believe that we have a good understanding of the factors that can lead to residual population stratification in the UK Biobank. First, not including PCs as covariates in the estimation of effect sizes. Second, extending the studied samples from the very homogeneous set of white British individuals to a wider range of samples with more diverse ancestry.

Conducting a meta-analysis per se should not be a factor that compromises stratification correction, as long as the principal components were computed on the whole sample. However, as the reviewer has pointed out, correcting for stratification within each cohort individually can lead to ineffective stratification correction, if cohort sizes are too small to allow PCA to capture the underlying population structure.

The distributed nature of a meta-analysis, and the difficulty of balancing transparency with data privacy concerns, make it easier for inconsistencies or mistakes to remain undetected and to have an impact on stratification correction. As we do not currently have access to cohort level data in GIANT, we are unable to comment on the causes of residual population stratification in the GIANT meta-analysis.

It would be possible to conduct an artificial meta-analysis in the UK Biobank and to induce residual stratification effects by computing PCs in very small cohorts or by omitting some PCs as covariates in some cohort. However, as there are several potential ways for stratification effects to enter into a meta-analysis, we would be unable to make strong conclusions about the origin of the observed differences between GIANT and the UK Biobank.

4) The Robinson et al., 2015 GWAS: Sib-based studies are done to avoid / minimize stratification effects and the Robinson 2015 data have been used as a proof of robustness in several previous studies. The fact that you find clear signs of stratification is sobering and one would like to know what has gone wrong. You may not currently have any explanation and this is fair enough. However, the discussion should be clearer and say upfront that results based on these data cannot be trusted until we understand the issues.

Again, repeating the above (the Major comments section). In the revised manuscript, we clarify that we agree with the conceptual approach of Robinson et al. but that the discrepancy is likely to be due to a technical error. We have in fact now confirmed this through correspondence with the authors of Robinson et al., and they are currently preparing a manuscript revisiting these analyses and correcting the technical issues. We emphasize that family-based effect size estimates computed in the UK Biobank following the Robinson et al. methodology behave as expected.

Minor Comments:a) You use 11 different summary statistics, with partly inconsistent naming strategy. I had to look up names in the methods part a number of times. I think this can be improved. Maybe even use the same names as Berg et al. where the summaries are identical.

We have made sure the names of the 11 summary statistics are consistent throughout the paper and figures. The main "UKB" dataset is referred to as "UKB Neale" in the supplemental figures to distinguish it from "UKB Neale new" and from the UKB LMM summary statistics "UKB Loh".

b) The switch from 1000 genomes to POPRES complicates comparison between figures. If there are advantages of POPRES, why not use it throughout? This holds, in particular, for the test of the latitudinal slope, which would be more convincing with many populations rather than just 4 from the 1000 genomes data.

We use 1000 genomes as it is publically available and several of the previous studies we analyze in this paper use 1000 genomes populations for claims of polygenic adaptation (Mathieson et al., Berg et al). The switch to POPRES for the test of latitudinal slope and overall overdispersion is to ensure that we do not see a nonsignificant P-value simply due to a lack of power in 1000 genomes. We realize this makes comparing Figures 1 and 4 difficult but we believe our paper is stronger for analyzing both datasets due to their different strengths and because both datasets have been used by previous studies of polygenic adaptation.

c) Figure 4: "The overdispersion signal disappeared entirely when the UK Biobank family based effect sizes were used": Is this due to the smaller sample size of the sib data or due to residual stratification issues in UKB? This could be tested using a sub-sample from UKB of the same size as the sib data.

Even without running another GWAS on a subset of the UK Biobank, we have some reasons to believe that the lack of overdispersion signal in the family-based estimates reflects a lack of power rather than residual stratification in the UK Biobank: Row 7 in Figure 4—figure supplement 1 shows that in the UK Biobank GWAS of white British samples without PC correction, no latitude signal is detectable, unless only genome-wide significant SNPs are used, which are least affected by stratification effects. We therefore think that the latitude signal at the genome-wide significant SNPs is likely real, rather than driven by stratification in the UK Biobank, as the stratification effects in UKB WB no PCs tend to go in the opposite direction. This leads us to believe that given enough power, a real overdispersion signal should probably be detected. However, we can’t help but remain agnostic as even for genome-wide significant SNPs (Figure 1—figure supplement 2B), polygenic scores for both modern and ancient individuals change when the main UKB summary statistics are used (WB ancestry controlling for 10 PCs) instead of GIANT. This shift, for example, for the hunter-gatherer (HG) polygenic score is troubling as we know that different European populations have variable amounts of ancient HG vs. EF vs. SP ancestry, and could reflect residual stratification in the UKB GWAS not captured by our PCs.

d) Figure 3 legend: "suggesting that tSDS shift at the gw-significant SNPs is not driven by population stratification": only true for stratification due to this particular axis.

We agree with the reviewer and have changed this sentence to “There is no significant difference in frequency in these two populations, suggesting that tSDS shift at the gw-significant SNPs is not driven by population stratification at least due to this particular axis.”

Additional data files and statistical comments:All necessary information is provided and the UKB sib data is on Dryad. I think the other newly generated GWAS data should go there, too.

We have placed all newly generated GWAS data on Dryad.